# The Role of Zinc in Modulating Acid-Sensing Ion Channel Function

**DOI:** 10.3390/biom13020229

**Published:** 2023-01-24

**Authors:** Amber W. Sun, Michelle H. Wu, Madhumathi Vijayalingam, Michael J. Wacker, Xiang-Ping Chu

**Affiliations:** Department of Biomedical Sciences, School of Medicine, University of Missouri-Kansas City, Kansas City, MO 64108, USA

**Keywords:** acid-sensing ion channels, zinc, function, neuron, modulation, neurological diseases

## Abstract

Acid-sensing ion channels (ASICs) are proton-gated, voltage-independent sodium channels widely expressed throughout the central and peripheral nervous systems. They are involved in synaptic plasticity, learning/memory, fear conditioning and pain. Zinc, an important trace metal in the body, contributes to numerous physiological functions, with neurotransmission being of note. Zinc has been implicated in the modulation of ASICs by binding to specific sites on these channels and exerting either stimulatory or inhibitory effects depending on the ASIC subtype. ASICs have been linked to several neurological and psychological disorders, such as Alzheimer’s disease, Parkinson’s disease, ischemic stroke, epilepsy and cocaine addiction. Different ASIC isoforms contribute to the persistence of each of these neurological and psychological disorders. It is critical to understand how various zinc concentrations can modulate specific ASIC subtypes and how zinc regulation of ASICs can contribute to neurological and psychological diseases. This review elucidates zinc’s structural interactions with ASICs and discusses the potential therapeutic implications zinc may have on neurological and psychological diseases through targeting ASICs.

## 1. Introduction

Acid-sensing ion channels (ASICs) are proton-gated, voltage-independent Na^+^ channels found predominantly in the central and peripheral nervous systems [1,2]. ASICs are part of the degenerin/epithelial sodium channel (DEG/ENaC) superfamily of amiloride-sensitive ion channels [3]. To date, at least six ASIC isoforms (ASIC1a, 1b, 2a, 2b, 3a, and 4) encoded on four genes (*ACCN1*–*ACCN4)* have been cloned [2,4]. These ASIC isoforms form homo- and heterotrimers with different physiological and pharmacological properties [3,4]. ASIC subunits, mainly those present in the brain, are activated by rapid drops in pH [3]. Transient pH drops lead to rapid desensitization of ASICs, producing a detectable current [3]. On the contrary, gradual acidification causes gradual desensitization. Thus, no detectable current is produced from ASICs with slow drops in pH [3]. Most ASIC subunits are located in the brain; however, ASIC3 is a calcium-insensitive channel that is predominantly found peripherally in the dorsal root ganglion (DRG) neurons and other locations in the peripheral nervous system (PNS) [2,3]. Moreover, ASIC4 is unlike other ASIC channels in that, despite its name, ASIC4 does not induce currents when protonated [3,5]. Rather than a protein channel, it is hypothesized to be a modulator, specifically downregulating ASIC1a and ASIC3 surface expression. In addition, ASIC4 is unique from other ASIC channels in that it is found intracellularly, mainly in endosome-related vacuoles [6,7].

In concurrence with the major physiological roles of ASICs, studies have indicated that the potentiation, desensitization, and inactivation of ASICs play a role in pathological processes such as brain ischemia [8], Parkinson’s disease [9], multiple sclerosis [10], and cocaine addiction [11]. It has been discovered that at both physiological and pathological levels, zinc serves as an important modulator of ASICs and can cause their activation or inhibition, depending on the specific ASIC isoform [4,12]. For example, the finger domain binds zinc, which has a potentiating effect on ASIC2a and an inhibiting effect on ASIC1b [13]. Beyond these findings, there are few studies reviewing the exact relationship of zinc with the different isoforms of ASICs. This review serves to elucidate zinc’s structural interactions with ASICs and discuss the potential therapeutic implications zinc may have on neurological and psychological diseases through targeting ASICs.

### 1.1. ASIC Structure

As a part of the DEG/ENaC superfamily, ASIC subunits are composed of intracellular NH_2_ and COOH terminals and two hydrophobic transmembrane domains (TMD1 and TMD2) separated by a large extracellular domain of approximately 370 residues [14,15,16,17]. The grossly viewed structure of an individual ASIC subunit’s extracellular domain resembles a “clenched fist” with wrist, palm, finger, knuckle, thumb, and β-ball domains [15,17]. The palm domain serves as the central structure within each extracellular ASIC subunit and has direct connections to the transmembrane domains (TMD1 and TMD2) and the thumb domain [17]. The knuckle domain and its attached finger domain lie superior to the palm domain [17]. The outer edges of the finger domain come in contact with the thumb domain, and altogether the palm, thumb, finger, and knuckle domains surround the small β-ball domain [17]. Between the β-ball, thumb, and finger domains lies a highly negatively charged cavity called the acidic pocket [17]. Within this acidic pocket are three pairs of carboxyl–carboxylate interactions between the side chains of aspartate or glutamate residues [17]. These carboxyl–carboxylate interactions are responsible for the ion and pH-sensing capabilities of the acidic pocket [15,17]. This extracellular “clenched fist” domain is rich in cysteine residues, and studies have indicated that mutating various cysteine residues in the extracellular domain of ASICs play a critical role in zinc’s potentiation and inhibition effects [15,17]. The zinc-binding site is located within the extracellular domain of the ASIC channel [4,12,18]. The specific extracellular domain zinc binds to varies on the ASIC subtype (see Figure 1). Zinc’s effects on the activation or inactivation of ASICs can be further regulated by extracellular residues, such as histidine and cysteine [4,12,18].

### 1.2. Zinc Physiology

Zinc is essential to the growth and development of living organisms, and after iron, zinc is the most abundant trace metal within the human body. The human body contains approximately 2–4 grams of zinc, and the majority of zinc is distributed within the testes, muscles, liver, and prostate [19,20,21]. Over 300 enzymes rely on zinc as their cofactor, and zinc plays an important structural role for various proteins [21]. Furthermore, zinc is an essential ion for neurotransmission and is widely distributed within the presynaptic vesicles [22]. Zinc plays a critical role in neurogenesis, as it controls the cell cycle, apoptosis, and the binding of DNA and several proteins [22]. In addition, zinc plays a significant role in inhibiting growth within the prostate gland [23]. For example, zinc inhibits the enzyme in the first step of the Krebs cycle, accumulating citrate within the prostate gland and preventing further downstream energy production [24,25]. Moreover, high levels of zinc induce mitochondrial apoptosis, dampening prostatic tissue’s growth and proliferation [25].

### 1.3. Zinc and Disease

The importance of zinc is underscored during the adverse effects that arise from a state of zinc deficiency. A lack of zinc in the body can lead to impaired immune function, growth retardation, delayed sexual maturation, poor wound healing, prostate cancer, and neurodegenerative disease [19,22,25,26,27]. Studies have indicated that men with prostate cancer have markedly decreased zinc levels by up to 80% compared to healthy controls. No study has reported a case of prostate cancer without decreased zinc levels [24,25].

Furthermore, zinc is involved in the pathogenesis of Alzheimer’s disease, ischemic stroke, traumatic brain injury, epilepsy, and Parkinson’s disease [26,27]. Most of these diseases are caused by zinc deficiency or intracellular zinc overload that disrupts multiple signaling pathways. A myriad of studies have been conducted to evaluate the concentration levels at which zinc develops neuroprotective or neurotoxic effects [27,28].

### 1.4. Zinc and Ion Channel Regulation

Ions cannot pass freely across cell membranes due to their charge; therefore, ion channels or transporters are required to transport ions between intra- and extracellular compartments [29]. Ion channels are mainly either voltage-gated or ligand-gated. Voltage-gated ion channels open and close based on changes in the membrane potential [30]. Ligand-gated ion channels rely on the release of molecules such as glutamate, glycine, acetylcholine, GABA, ATP, and serotonin [31]. These messengers selectively open cationic or anionic channels to depolarize or hyperpolarize the cell, respectively [31]. If a channel allows for the influx of sodium or calcium, it is classified as excitatory. Likewise, if a channel allows for the influx of chloride ions or efflux of potassium, it is classified as inhibitory. Zinc has been studied for the regulation or modulation of several ion channels, including the K^+^ channel, Ca^2+^ channel, *N*-methyl-*D*-aspartate (NMDA) receptors, α-amino-3-hydroxy-5-methyl-4-isoxazole propionic acid (AMPA) receptors, kainate receptors, transient receptor potential (TRP) channels, and γ-aminobutyric acid (GABA) receptors [31,32].

K^+^ channels selectively transport potassium from inside the cell to the extracellular space [33]. There are four main classes of K^+^ channels: calcium-activated K^+^ channels, inward-rectifying K^+^ channels, tandem pore domain K^+^ channels, and voltage-gated K^+^ channels [33]. Each K^+^ channel is divided into a pore-forming and regulatory domains [33]. The pore-forming domain allows for the passage of K^+^ ions, and its structure is conserved among the different types of K^+^ channels [33]. The regulatory domain varies in structure depending on the K^+^ channel type [33]. Potassium channels are widely expressed in the peripheral and central nervous system cell membranes and play a major role in multiple cellular physiological processes as they control the resting membrane potential, repolarization rate of action potentials, and spike frequency adaptation [33]. Therefore, potassium channel dysfunction is associated with multiple neurological disorders such as epilepsy [34], Huntington’s disease [35], and Parkinson’s disease [36,37]. At micromolar concentrations, zinc is a negative modulator for most K^+^ channels [32]. The exceptions are that zinc positively modulates TWIK-related potassium channels subtype 2 (TREK-2) at an EC_50_ of 659 µM and voltage-gated potassium channels subfamily Q member 5 (KCNQ5) at an EC_50_ of 22 µM [32]. Furthermore, zinc activates Slo1 K^+^ channels at an EC_50_ of 34 µM [32].

There are two types of Ca^2+^ channels: high voltage-activated (Ca_V_1.1–1.4, Ca_V_2.1–2.3) and low voltage-activated (Ca_V_3.1–3.3) channels [38]. These voltage-gated calcium channels (VGCC) open for the entry of Ca^2+^ into the cell in response to membrane depolarization [38]. Zinc serves as a negative modulator for all calcium channels [32]. Ca_V_1.1–1.4 channels conduct L-type calcium currents, distinguished by slow voltage-dependent inactivation [38]. The L-type Ca^2+^ channels are mainly expressed in smooth muscle cells [32]. These channels are involved in initiating contraction, hormone secretion, and local calcium signaling to gene transcription [38]. Ca_V_1.1 has a zinc IC_50_ of 11 µM or 18 µM, and Ca_V_1.2 has a zinc IC_50_ of 34 µM [32]. Ca_V_2.1–2.3 channels conduct P/Q-, N-, and R-type calcium currents, respectively [38]. These currents contain faster voltage-dependent inactivation and are located in neurons [38]. Ca_V_2.1 has a zinc IC_50_ of 110 µM, Ca_V_2.2 has an IC_50_ of 98 µM, and Ca_V_2.3 has an IC_50_ of 32 µM [31]. Ca_V_3.1–3.3 channels conduct T-type calcium currents, which are activated at negative membrane potentials [38]. T-type calcium currents also have fast deactivation upon repolarization and fast voltage-dependent inactivation during sustained depolarizations [38]. Ca_V_3.1–3.3 channels are prominent within cardiac myocytes in the sino-atrial node and neurons within the thalamus [38]. They have zinc IC_50_s of 82, 0.8, and 159 µM or 196, 24, and 152 µM for Ca_v_3.1, Ca_v_3.2, and Ca_v_3.3, respectively [32]. Mutations in calcium channels are responsible for neuropsychiatric diseases, migraines, hypertension, heart failure, and chronic pain syndromes [39].

Glutamatergic neurons are involved in important roles in the CNS, such as learning/memory and synaptic plasticity [40]. Glutamatergic neurotransmission is primarily modulated through ionotropic and metabotropic glutamate receptors [40]. The ionotropic glutamate receptors are ligand-gated ion channels that are permeable to Na^+^ and K^+^. Certain ionotropic glutamate receptors have Ca^2+^ permeability, such as GluN2-containing subunits [40]. There are three groups of ionotropic glutamate receptors: *N*-methyl-*D*-aspartate (NMDA) receptors, α-amino-3-hydroxy-5-methyl-4-isoxazolepropionic acid (AMPA) receptors, and kainate receptors [40]. At normal physiological conditions, NMDA receptors are blocked by Mg^2+^. To function properly, NMDA receptors first require membrane depolarization via an influx of Na^+^ from AMPA receptors to remove this Mg^2+^ block and allow subsequent ion permeability [40]. Once the Mg^2+^ block is removed, NMDA receptors allow the influx of Na^+^ and Ca^2+^ [40]. This influx of Ca^2+^ activates intracellular mechanisms that lead to the phosphorylation and subsequent upregulation of AMPA receptors [40]. AMPA and NMDA receptors play a major role in synaptic plasticity and are involved in pathological processes such as Alzheimer’s disease [41]. Kainate receptors and AMPA receptors are inhibited by zinc [31,32]. The regulation of zinc on these ionotropic glutamate receptors heavily depends on the subunit composition [31,32]. For example, the co-expression of GluN1 and GluN2A subunits in HEK293 cells increased the negative zinc modulation by 1000-fold compared to the co-expression of GluN1 and GluN2B subunits in HEK293 cells [31]. Ionotropic glutamate receptors, particularly NMDA and AMPA receptors, have been linked to neurological disorders such as Alzheimer’s disease, ischemic stroke and schizophrenia [41,42,43].

TRP channels are a large family of channels that are activated by a variety of sensory stimuli [44]. Out of the 28 members, there are six TRP subfamilies: canonical (TRPC), vanilloid (TRPV), melastatin (TRPM), polycystin (TRPP), mucolipin (TRPML), and ankyrin (TRPA) [44]. These channels are integral in evaluating environmental stimuli and act as signal transducers via altering the membrane potential and intracellular Ca^2+^ levels [44]. TRP channels are involved in a large number of physiological processes and are potential therapeutic targets. For example, TRPA contains a single chemo-nociceptor that may be a potential analgesic target, and TRPP is involved in autosomal dominant polycystic kidney disease [44]. Zinc serves as a negative modulator for TRPM2 and TRPM5 [31,32]. When TRPM2 is activated by ADP ribose, zinc negatively modulates the channel because of inhibition of the channel above levels of 30 µM of extracellular zinc [31,32]. When TRPM5 is activated by 500 nM of intracellular Ca^2+^, it has a zinc IC_50_ of 4.3 µM [31,32].

GABA is a fast inhibitory neurotransmitter predominant within the CNS and acts on GABA_A_ and GABA_B_ receptors [45]. A dysfunction with these receptors can lead to a wide array of neurological problems [45]. GABA_A_ receptors are the target for multiple therapeutic drugs, including those for epilepsy, anxiety, insomnia, and panic disorder [45]. GABA_A_ receptors are composed of 19 subunits, with the predominant presynaptic isoform composed of two α1 subunits, two β2 subunits and one γ2 subunit [45]. GABA receptors are negatively modulated by zinc [32]. The GABAρ1 subunit has a zinc IC_50_ of 22 µM or 20 µM, the GABAα1β2γ2 subunit has a zinc IC_50_ of 441 µM, and GABA_A_ has a zinc IC_50_ of 7 µM [32].

Despite the research into zinc and other membrane channels, few studies or reviews have linked zinc’s pathological role in these neurodegenerative disorders with its effects on ASICs [13]. Both zinc and ASICs are involved in the pathophysiology of multiple neurological and psychological disorders. Depending on the ASIC subtype and composition, zinc can serve as a potentiator or inhibitor of that channel by binding to the ASIC extracellular domain. Further study of the connection between zinc and ASICs may reveal a critical point for the therapeutic treatment of several of these “incurable” neurodegenerative disorders. Thus, the next section of the review details the relationship between zinc and ASICs and how these two substances correlate with several neurological and psychological diseases.

## 2. Zinc’s Effects on Different Types of ASICs

### 2.1. Zinc and ASIC1a

Although ASIC1a and ASIC1b are transcribed from the *ACCN2* gene [2,16], their function and location vary significantly. ASIC1a is widely expressed in the CNS and PNS. Centrally located homotrimeric ASIC1a plays a vital role in synaptic plasticity, learning, and memory [3]. ASIC1a also plays a role in fear and anxiety [3]. Unlike the other ASIC channels, ASIC1a is uniquely permeable to Ca^2+^ and is heavily involved in acidosis-induced injuries such as ischemic brain injury [46,47,48,49,50]. Though their calcium permeability is poor, activation of ASIC1a channels may induce damage through secondary mechanisms that increase intracellular Ca^2+^ levels, such as the activation of voltage-gated Ca^2+^ channels and intracellular storage release [50]. Zinc chelation potentiates ASIC1a-mediated currents, consequently increasing intracellular calcium levels and inducing membrane depolarization [46].

Our laboratory’s previous studies have shown that zinc has a high-affinity binding site on the lysine-133 residue in the extracellular domain of ASIC1a channels, where it exhibits inhibitory behavior. Consequently, mutations of K133 rendered zinc’s function to be obsolete [46]. The presence of 0.3 μM of zinc displayed reversible inhibition of ASIC1a [50]. The effects of *N,N,N′,N′*-tetrakis(2-pyridinylmethyl)-1,2-ethanediamine (TPEN), a high-affinity zinc chelator, on closed-state ASIC1a revealed current potentiation with an EC_50_ value of 2 μM in mouse cortical neurons [46]. The maximum potentiation induced by TPEN can be achieved in cells within 2–4 min of perfusion, and the effect is reversible following a washout of TPEN. Adding TPEN alone in the solution does not trigger any current in pH 7.4 solution, and co-applicating TPEN to pH 6.5 solution for 10 seconds does not induce any significant potentiation. Another study revealed that the co-application of 300 μM of zinc to *Xenopus* oocytes displayed slight inhibition of the ASIC1a current that was not significant [18], suggesting that zinc only has significant inhibitory effects on the closed state of ASIC1a channels.

Zinc-mediated inhibition is pH- and dose-dependent based on zinc chelation using 10 μM of TPEN in various concentrations of buffered free zinc ions with an IC_50_ value of 14 nM in mouse cortical neurons [46]. Consequently, a zinc dose-inhibition curve yielded an IC_50_ value of 7.0 ± 0.35 nM for buffered zinc solutions. An increase of zinc concentration up to 30 μM does not display additional zinc-mediated inhibition [46].

### 2.2. Zinc and ASIC1b

ASIC1b and ASIC3 channels normally possess a transient current followed by a sustained current, which contributes to prolonged acidosis and pain sensation [12,51,52]. Zinc binds to the extracellular domain of ASIC1b channels and displays inhibitory mechanisms [12,51]. Zinc inhibits the peak amplitude in both ASIC1b and ASIC3 channels, but unlike ASIC1b, zinc inhibits both the peak and sustained component in ASIC3, thereby indicating that zinc has differing mechanisms on ASIC1b and ASIC3 [12,51,52].

Zinc concentrations of 1 or 3 μM harbor no significant inhibition on ASIC1b channels [12,51] while concentrations of 10, 30, 100, or 300 μM exhibit a profound and concentration-dependent inhibitory effect of the peak amplitude [12,51], revealing that zinc has a low-affinity binding site of ASIC1b channels. Furthermore, pretreatment of zinc possesses inhibitory behavior with an IC_50_ of 36.5 ± 1.5 μM while co-application harbors no effects on ASIC1b currents [12,51]. Additionally, pH activation, steady-state desensitization, and extracellular calcium concentrations harbor no effect on zinc-mediated inhibition of ASIC1b channels, indicating non-competitive processes [12,51]. Therefore, zinc reveals a strong pH- and calcium-independent inhibitory effect on the peak component of ASIC1b channels in closed states only.

Studies from our laboratory have shown that mutation of the extracellular cysteine-149 residue in mouse ASIC1b did not reveal any zinc-mediated inhibition, indicating a lack of zinc-binding site in rat ASIC1b when cysteine 149 residue was replaced [51]. Patch-clamp recordings of zinc and human ASIC1b with a mutation of the extracellular cysteine-196 residue demonstrated no inhibition of the current due to the lack of zinc binding [12]. Consequently, mutating other cysteine or non-cysteine residues on ASIC1b channels exhibited a reduction of ASIC current in peak amplitude by a drop in pH [12], revealing that the extracellular cysteine-196 residue of human ASIC1b is the low-affinity binding site of zinc responsible for mediating inhibitory mechanisms [12].

### 2.3. Zinc and ASIC1a/3

ASIC channels can form homomeric and heteromeric channels [4,46]. ASIC1a is predominantly located centrally, whereas ASIC3 is primarily expressed in the peripheral sensory neurons. The coexistence of the ASIC1a/3 heterodimer is currently discovered to be located mostly in skeletal muscle and plays a role in pain management [53,54]. Recently, we found that systematically mutated histidine residues 72 and 73 in both ASIC1a and ASIC3 and histidine residue 83 in ASIC3 were responsible for the dual effects of zinc on heteromeric hamster ovary ASIC1a/3 channels [4].

Co-application of zinc dose-dependently potentiated the peak and sustained component of ASIC1a/3 channels. Concentrations between 1 and 100 μM displayed an EC_50_ of 26 μM whereas concentrations between 100 and 1000 μM displayed an EC_50_ of 343 μM [4]. Overall, zinc has a low-affinity binding site on ASIC1a/3 that mediates potentiation of both the peak and sustained components of open-state ASIC1a/3 heteromeric channels in a dose- and pH-dependent manner.

Pretreatment with zinc between 3 to 100 μM exerted the same potentiation as co-application [4]. Concentrations between 1 and 100 μM exhibited an EC_50_ of 24 μM, and concentrations between 100 and 250 μM exhibited an EC_50_ of 128 μM. In contrast, concentrations above 250 μM exerted profound inhibition on the peak amplitude with an IC_50_ of 306 μM [4]. Taken together, zinc displays dual effects on the closed state of ASIC1a/3 channels in a dose- and pH-dependent manner [4].

### 2.4. Zinc and ASIC1a/2b

Homomeric ASIC2b channels do not produce currents independently, but ASIC2b, associated with other ASIC subunits, can form functional heteromeric ASIC channels [55]. The ASIC2b subunit enables the heteromeric channel ASIC1a/2b to harbor unique channel properties different from homomeric ASIC1a [55]. ASIC1a/2b undergoes steady-state desensitization at more basic pH values (pH 7.4) than other ASIC channels [55]. In *Xenopus* oocytes, co-application of 300 μM of zinc on ASIC1a/2b heteromers and ASIC1a homomers displays reversible inhibition. This is in contrast to ASIC1a/2a, which demonstrates profound potentiation. However, zinc modulation has reduced effects in the presence of ASIC2b with ASIC1a compared to ASIC1a homomers [55]. The same study also utilized 10 μM TPEN and assessed that nanomolar chelating concentrations of zinc enhance the ASIC1a/2b current amplitude. In fact, ASIC1a, ASIC1a/2a, and ASIC1a/2b are all inhibited by low nanomolar concentrations of zinc [55]. These studies monitored all currents with zinc modulation after the voltage ramp, indicating that these channels were assessed in an open state. The exact zinc-binding site of ASIC1a/2b is unknown, although it is strongly hypothesized that it is located on the ASIC2 subunit based on the differing current modulation of zinc when comparing homomeric ASIC1a and ASIC1a/2b [55]. Further research is required to identify the exact binding site for zinc modulation.

### 2.5. Zinc and ASIC1a/2a

Zinc has a dual dose-dependent effect on ASIC1a/2a heteromers. At high micromolar concentrations (100–300 μM), zinc binds to the ASIC1a/2a channels with low affinity and potentiates the effect of the ASIC1a/2a heteromers as it does with ASIC2a [18,46]. Zinc potentiates the ASIC1a/2a current at an EC_50_ of 111 μM upon co-application with an acidic pH [18]. Zinc cannot potentiate the channel unless the ASIC1a/2a heteromer is in an open state. Normally, ASIC1a/2a requires a more acidic pH to be activated in the CNS. However, the potentiation of ASIC1a/2a by zinc at high micromolar concentrations leftward shifts the pH dependence to a pH closer to physiological pH. Rather than becoming activated at a pH_0.5_ of 5.5 as the channel usually does, the pH_0.5_ for ASIC1a/2a channels activated by zinc is higher, at 6.0 [18]. The greatest potentiation by zinc was between a pH of 6.9 and 6.3 [46]. These heteromers increase the Hill coefficient by shifting the activation curve to the left, demonstrating that one ASIC2a subunit is enough for Zn^2+^ to potentiate the heteromer [18]. In homomeric ASIC2a channels, mutation of either the histidine residue H162 or H339 to alanine inhibits zinc coactivation of the channel [18]. Conversely, zinc coactivation of the heteromeric ASIC1a/2a channel is inhibited by H339A mutations but not by H162A mutations. This indicates a difference between the zinc binding sites of the ASIC2a containing homomeric and heteromeric channels that can alter the capability of zinc as a potentiator [14]. An increasing number of 2a subunits in comparison to 1a subunits also does not increase its potentiation by Zn^2+^. Unexpectedly, the ASIC1a-2a-1a concatemer with two 1a subunits had greater potentiation by zinc than the ASIC1a-2a-2a concatemer [56]. This suggests that the cooperation of the three subunits contributes to its ability to be potentiated by Zn^2+^ rather than simply the presence of ASIC2a.

At nanomolar concentrations, Zn^2+^ inhibits the ASIC1a/2a heteromer as it does with ASIC1a [46]. Chelation of zinc by TPEN potentiates the current through ASIC1a/2a in a dose-dependent manner [46]. The IC_50_ value for inhibition of ASIC1a/2a by zinc is 10 nM [46]. While the absence of ASIC2a did not impact the inhibition of zinc, the absence of the ASIC1a subunit did eliminate the inhibition of zinc, confirming that the presence of ASIC1a is necessary for the inhibitory effects of zinc [46]. The high-affinity site K133 located extracellularly on the ASIC1a subunit is highly indicated in the binding of zinc to the channel. This positively charged lysine residue could be involved in the inhibitory effect of zinc at nanomolar concentrations on ASIC1a/2a heteromers since ASIC1a mediates zinc inhibition [46]. However, the exact binding site for the inhibitory action of zinc on the ASIC1a/2a channel is currently unknown.

### 2.6. Zinc and ASIC2a

Unlike ASIC1a channels, high micromolar concentrations of zinc (e.g., 100 or 300 μM) potentiate ASIC2a and ASIC2a-containing channels [18,46]. Upon mutation of the extracellular His-162 (H162) and His-339 (H339) residues to alanine, zinc was no longer able to potentiate ASIC2a-containing channels [18]. Specifically, zinc binds with low affinity to the H339 and H162 residues at the interface between the upper palm, finger, and ball domains [18]. This is different from heteromeric ASIC2a-containing channels in which H339 is necessary, but H162 has either moderate or no effect on zinc sensitivity, indicating a difference in binding sites between the homomers and heteromers [18]. ASIC2a is potentiated by zinc at an EC_50_ of 120 μM upon co-application with an acidic pH [18]. Thus, zinc potentiates the ASIC2a homomer when the channel is in an open state. Homomeric ASIC2a normally has low acid sensitivity and cannot be activated at a pH between 7.4 and 5.5. It requires a pH of 4.5 to activate a large current [18,46]. However, the potentiation of ASIC2a by zinc occurs at a much lower concentration of H^+^ between a pH of 6.9 and 5.0 since zinc induces an alkaline shift of the pH dependence [18]. The pH sensitivity of ASIC2a is dependent on the extracellular His-72 residue found immediately after the first transmembrane domain and is abolished if this is mutated [18].

### 2.7. Zinc and ASIC2a/3

Another function of ASIC2a is to increase the conductance sensitivity of the cell membrane to protons by increasing the expression of ASIC3 at the cell surface and assembling heteromers with ASIC3 [56,57]. ASIC2a/3, like the previous ASIC2a-containing channels, is also potentiated by zinc [18]. The potentiation of this channel by micromolar concentrations of zinc mimics that of ASIC1a/2a, with the greatest potentiation being at a pH of 6 [18]. Zinc acts on the ASIC2a/3 channel in an open state, requiring co-application with an acidic pH for potentiation to occur [18]. Similar to ASIC1a/2a, potentiation by zinc decreased with increasing extracellular acidity [18]. While the effect of zinc on the potentiation of ASIC1a/2a was unchanged with a mutation of H162, for ASIC2a/3 channels, both H162 and H339 mutations decreased the effects of zinc [18]. This indicates that zinc likely binds with low affinity to both the H162 and H339 residues to potentiate the heteromeric ASIC2a/3. However, the H339 residue is likely more significant because an H339 mutation had a greater zinc potentiation reduction than the H162 mutation [18].

### 2.8. Zinc and ASIC3

Zinc binds to the extracellular domain of ASIC3 channels and exhibits inhibitory behavior [52]. Similar to ASIC1b, ASIC3 channels possess a transient current followed by a sustained current, thereby prolonging acidosis and further contributing to pain perception [54]. Nanomolar concentrations of zinc harbor no involvement in ASIC3 channels, while micromolar concentrations exhibit inhibitory behavior on both the transient and sustained components of ASIC3 currents, indicating that zinc has a low-affinity binding site on ASIC3 channels [52]. Additionally, our studies have shown that the co-application of zinc has no effects on ASIC3 currents, whereas pretreatment of zinc displays dose-dependent inhibition with an IC_50_ of 61 ± 3.2 µM [52]. Zinc-mediated inhibition occurs rapidly and at a narrow concentration range between 30 to 300 µM [52]. Thus, zinc reveals a strong inhibitory effect on ASIC3 channels in closed states within a narrow micromolar concentration range.

Administration of intracellular zinc does not diminish zinc-mediated inhibition of ASIC3 currents, confirming that the zinc-binding site is located outside the cell [52]. When administering zinc on ASIC3 channels with calcium concentrations at 2, 5, or 10 mM, the degree of zinc inhibition remains unchanged, concluding that zinc and calcium do not share the same binding site on ASIC3 channels [52]. Therefore, zinc-mediated inhibition is calcium-independent for this particular isoform. When comparing zinc inhibition at various pH values, the percent inhibition does not deviate significantly, suggesting that this inhibition is pH-independent [52]. With consistent inhibitory behavior and pH activation acuity, zinc may be a crucial ASIC3 channel regulator in pathophysiological conditions associated with pH changes, such as epilepsy [58,59], myocardial ischemia [60], rheumatoid arthritis [61,62], Alzheimer’s disease [63], and trauma [64].

While zinc commonly binds to histidine or cysteine residues in other ASIC channels, modification of these two amino acid residues on ASIC3 did not affect zinc inhibition, suggesting that zinc binds to a site unrelated to histidine or cysteine [52]. Further examination of other extracellular residues like glutamate may help to identify the specific zinc-binding site of the ASIC3 channels. Table 1 shows zinc’s effects on different type of ASICs.

## 3. Zinc Regulation of ASICs in Neurological and Psychological Diseases

### 3.1. ASIC1a

#### 3.1.1. Epilepsy

Epilepsy is a neurological disorder characterized by abnormal, excessive, or synchronized neuronal activity [65]. Various studies reveal the significance of zinc homeostasis in seizures and epilepsy. Zinc is necessary for proper neural signaling, whereas zinc dyshomeostasis leads to an improper balance of neural excitation and inhibition, resulting in seizures [20]. Increased zinc serves as a protective tool in preventing ASIC1a- and NMDA-mediated excitotoxicity in neuropathological conditions like epilepsy [59,66]. Activating ASIC1a channels from decreased pH levels in the brain leads to acidosis-mediated neurological damage [67]. The zinc-mediated inhibition of ASIC1a channels may reduce acidosis and thus prevent brain injury from seizures [67].

Further research on administering zinc as a therapy to epileptic patients is required to document the specific relationship between zinc and epilepsy. Notably, increased zinc concentrations beyond physiological concentrations were found to be toxic due to their entry into neurons [34,68]; yet another study found that a moderate increase of zinc in the extracellular space is neuroprotective against pathological conditions with severe acidosis. The corresponding study used HEK 293 cells, so future research is needed on primary neurons to study the neuroprotection of different concentrations of extracellular zinc [67].

#### 3.1.2. Migraines

The pathophysiology of migraines is poorly understood, although one hypothesis suggests extracellular acidification [69]. Calcium ions play a vital role in the human body and can contribute to non-mitochondrial reactive oxygen species (ROS) production [70] and induce acidosis-mediated injury [46]. Various studies reveal a correlation between migraine headaches and zinc deficiency [71]. A 2020 randomized 8-week clinical trial revealed that 220 mg of zinc sulfate per day reduced the frequency of migraine attacks in comparison to the placebo group. However, other factors such as headache severity, migraine duration, and presence of auras were not affected by zinc supplementation [72]. Similarly, a 2021 randomized 12-week clinical trial revealed that zinc glucose supplementation not only significantly reduced the frequency but also the periods and severity of migraine attacks in comparison to the control group [73]. Taken together, zinc supplementation has an overall positive impact on migraine attacks by reducing the frequency and severity. ASIC1a is vital for normal brain function, though due to its calcium permeability, excessive ASIC1a signaling can contribute to acidosis-mediated injury [50] and cortical spreading depression in migraines [71]. Administration of the inhibitors mambalgin-1 and amiloride to ASIC1a channels revealed significant efficacy as acute and prophylactic treatment options for migraines [71]. Most therapeutic options for chronic migraines are notoriously difficult and commonly fail [74,75]. Further research is necessary to analyze the potential therapeutic effects of zinc administration and chronic migraines. Thus, zinc supplementation might reveal a potential therapeutic option for the treatment of acute migraine attacks due to its inhibitory modulator on ASIC1a.

#### 3.1.3. Alzheimer’s Disease

Alzheimer’s disease (AD) is a progressive neurodegenerative disorder and is one of the leading causes of dementia [76]. It is characterized by advanced cognitive impairment associated with behavioral changes, memory loss, and learning and orientation complications [76,77]. Diagnostic histopathologic features of AD include amyloid plaques aggregated by β-amyloid (Aβ) peptides and neuronal fibrillary tangles (NFTs) by hyperphosphorylated tau protein in the brain [78,79].

The imbalance of zinc in the brain is one of the pathological features of AD. Zinc concentrations are elevated in specific regions of the brain affected by AD, which may be a result of the Aβ peptides capturing zinc ions. Additionally, zinc directly promotes the aggregation of Aβ peptides and tau hyperphosphorylation, thereby exacerbating the advancement of AD [80,81,82,83,84]. Furthermore, the quantity of zinc transporters decreases as the disease progresses, correlating to increased disease severity and cognitive impairment. It is unclear whether zinc concentrations or zinc transporters are the ultimate cause of AD [84,85].

ASIC1a channels may play a role in the pathogenesis of AD through their involvement in the Aβ-mediated effect on metabotropic glutamate (mGlu) receptor-dependent transmission. Consequently, utilizing the ASIC1a-selective inhibitor, psalmotoxin-1, restored the intrinsic excitability of mGlu in the hippocampus, revealing that ASIC1a channels play a role in the Aβ-related depolarizing response and long-term depression [77]. Taken together, the relationship between ASIC1a channels and mGlu potentially suggests their significant role in the pathogenesis of AD.

Because both zinc and ASIC1a channels play a role in the pathogenesis of AD, targeting zinc and ASIC1a may have therapeutic potential for AD. Various studies revealed potential therapeutic effects of zinc and copper balance on the early stages of AD [86], unrelated to ASIC channels. Further research is required to analyze the relationship between zinc and ASIC1a channels in patients with AD.

#### 3.1.4. Parkinson’s Disease

Parkinson’s disease (PD) is a neurodegenerative disorder characterized by Lewy body inclusions and the degeneration of the dopaminergic neurons in the substantia nigra pars compacta [87], leading to dopamine deficiency in the striatal pathway and, ultimately, basal ganglia deterioration [88,89]. Clinical presentation of PD includes olfactory dysfunction, tremor, cogwheel rigidity, cognitive impairment, and more [87].

It is uncertain how zinc dyshomeostasis and PD are related. Some studies have found lower serum zinc levels in PD patients [90,91], while others have found higher serum zinc levels. Studies reveal that methamphetamine causes dopaminergic cell death by generating reactive oxygen species and increasing the total amount of α-synuclein, a key element of Lewy bodies [85]. Zinc pretreatment reverses the aforementioned phenomena by increasing metallothionein expression in vitro, attenuating the accumulation of ROS in neurons [92,93]. By pretreating the cells with 50 μM of zinc chloride, methamphetamine-induced expression of α-synuclein was significantly reduced [92]. This observation supports the potential relationship between low zinc levels and α-synuclein production in PD. Contrastingly, numerous studies have demonstrated the detrimental effects of an aberrant accumulation of zinc in the substantia nigra and striatum, leading to dopaminergic neuronal cell death [88,94]. These findings are reinforced by the observation of 1-methyl- 4-phenyl-1, 2, 3, 6-tetrahydropyridine (MPTP)-induced neuronal cell death in mice when zinc is pretreated [85]. In principle, these research findings imply that zinc dyshomeostasis may harbor dual involvement in the pathogenesis of PD contingent on zinc-modulated signaling pathways at specific stages of the disease.

Aberrant excess of neural inflammation and lactic acidosis contribute to neurodegeneration in PD [87,95]. Observation of lactic acidosis in the animal model of PD reveals that ASIC1a may also play a role in dopaminergic neuronal cell death. Amiloride (non-selective ASIC inhibitor) and psalmotoxin-1 (selective inhibitor of ASIC1a) were revealed to attenuate neurodegeneration in the substantia nigra pars compacta [95]. This observation is supported by the conclusion that ASIC1a may play a significant role in the pathophysiology of PD [77], either by mutations in the Parkin gene associated with the autosomal recessive juvenile-onset of PD [96] or by the absence of the Parkin gene, which promotes hippocampal ASIC1a currents [97]. The role of ASIC1a as a therapeutic target for PD has not been the subject of many research investigations.

The significance of zinc and ASIC1a channels as a potential treatment of PD demands further investigation. Zinc-mediated inhibition of ASIC1a channels [46] is a probable therapeutic target to prevent neurodegeneration, but future research is required to detail zinc’s precise effects on ASIC1a channels and how it impacts PD.

#### 3.1.5. Depression

Major depressive disorder (MDD) is a prevalent mental illness with unclear etiology and poor effective therapies [98,99]. A key contributing factor to depression may be alterations to serotonin levels in the brain [99]. The glutamatergic theory, which postulates that depression results from an imbalance between the excitatory effect of glutamate and the inhibitory action of γ-aminobutyric acid (GABA), is the widely accepted approach to understanding the pathophysiology of depression. However, the glutamatergic theory of depression does not account for the large spectrum of symptoms observed in MDD [99]. Consequently, there is potential significance between zinc and its receptor, GPR39, that bridges the gaps in understanding depression. Therefore, it is crucial to investigate potential interactions between the brain’s monoaminergic, glutamatergic, and zincergic systems.

Zinc can act as an inhibitory neuromodulator of NMDA channels, a major pharmacotherapeutic target in depression, [100] or as a neurotransmitter [101]. Neural transmission is disrupted in zinc deficiency, clinically manifesting as cognitive, emotional, and behavioral impairment [102]. In rats, low levels of zinc cause elevated cortisol, enhancing the hypothalamic–pituitary–adrenal (HPA) axis and ultimately facilitating the pathogenesis of depression [103]. GPR39 also plays a significant role relevant to cognition, emotions, and memory processing [101,102]. When bound to zinc, GPR39 is hypothesized to participate in serotonin synthesis [104], serotonin receptor signaling [105], and higher brain-derived neurotrophic factor (BDNF) [106]. Consequently, GPR39 knock-out mice are resistant to traditional antidepressants [104]. Zinc is also hypothesized to directly affect serotonin signaling through agonistic binding to the 5-HT1A receptor and antagonistic binding to the 5-HT7. Likewise, zinc transporters play a role in depression. Mice with absent zinc transporter-3 exhibit reduced proliferating progenitor neurons [107] and decreased hippocampal volume [108], suggesting that alterations to zinc transporter-3 contribute to the pathogenesis of depression. The impacts of zinc are comparable to that of typical antidepressants, and prior research has demonstrated that antidepressant treatment of depression restores low BDNF levels [109].

Fear, addiction, and depression are attributed to ASIC channels [110,111]. ASIC1a channels located in the amygdala are hypothesized to play a significant role in depression [110,111,112]. ASIC1a channel disturbance is theorized to disrupt the fear circuit, leading to deficits in fear-related behavior [112]. A 2009 study investigated the antidepressant-like effects when disrupting ASIC1a channels in mice. Using the forced swim test and tail suspension test, mice with absent ASIC1a channels displayed antidepressant-like findings compared to mice with normal ASIC1a. Administration of PcTx1 and amiloride also produced antidepressant-like effects. Overall, findings from this study reveal that ASIC1a contributes to depression in mice [110]. Surprisingly, a 2017 study revealed normal nucleus accumbens ASIC1a expression in knock-out mice with absent ASIC1a alleles. The same mice also demonstrated a normal forced swim test. The discrepancy may be due to the newer study using SynAsic1a KO mice generated by floxed ASIC1a alleles disrupted by Cre recombinase driven by the neuron-specific synapsin I promoter, whereas the 2009 study used ASIC1a^−/−^ mice. Compared to ASIC1a^−/−^ mice, SynAsic1a KO mice do not display identical behavioral changes but have similar deficits in fear conditioning. Furthermore, not all neurons had disrupted ASIC1a expression in SynAsic1a KO mice [109]. Therefore, ASIC1a channels impact fear-related behaviors in mice, but further research is required to assess the specific relationship between ASIC1a and depression. With zinc intrinsically harboring antidepressant effects, targeting ASIC1a channels using zinc-mediated inhibition may be another potential antidepressant. Future investigation is warranted to identify a potential correlation between ASIC1a channels, zinc, and depression in humans.

#### 3.1.6. Stroke

Ischemic stroke is characterized as a thrombo-inflammatory condition that induces a pro-inflammatory state at the site of vascular injury, consequently compromising the blood–brain barrier (BBB) and inducing neuronal cell death [113,114]. Under physiological conditions, zinc and other divalent cation transport are maintained by the BBB [115]. Some research utilizing animal models of global ischemia in cortical neurons reveals that zinc accumulation drives the progression of brain infarction. Additionally, pathological zinc concentrations in the synaptic cleft of ischemic neurons also elicit cell death, revealing that zinc toxicity may be an independent risk factor for ischemic stroke [115,116]. Consequently, chelating zinc using EDTA in rats with ischemia revealed neuroprotection through enhanced cognitive function and deterred apoptosis of ischemic cells [116]. Surprisingly, other studies have revealed that zinc administration in rats with cerebral ischemia guards the hippocampus against neuronal injury during the reperfusion phase [115,117].

Furthermore, patients with zinc deficiency have a greater likelihood of ischemic strokes and an enhanced rehabilitation of neurological deficits with zinc supplementation [115,118]. These contradictory findings suggest that zinc plays a variety of functions in both the early and late stages of ischemic strokes. The laboratory settings may also substantially influence the beneficial or detrimental effects of zinc addressed above.

ASIC1a channels are suggested to play a role in the progression of ischemic strokes due to their activation during hypoxia [119,120]. Lactic acid production from increased anaerobic glycolysis promotes an acidic pH in the brain, thereby activating ASIC1a and ASIC1a-containing channels [119]. ASIC1a-mediated neuronal ischemic injury is further enhanced by concurrent induction of other elements such as Ca^2+^/calmodulin kinase II and NMDA receptors [121]. Consequently, the administration of amiloride, a high-affinity inhibitor of ASIC1a and most other ASIC subtypes, demonstrated a reduced cerebral ischemic cell injury [120]. Taken together, the inhibition of ASIC1a channels may have significant therapeutic potential in ischemic strokes. Within the current literature, the effect of zinc on ischemic stroke is inconclusive, as different studies have seen the neuroprotective effects of both zinc chelation and administration. However, further research on the effect of zinc chelation and administration in the early versus late stages of ischemic stroke may ascertain whether the neuroprotective effects of zinc chelation are time-dependent and/or superior to the effects of zinc administration.

#### 3.1.7. Cocaine Addiction

Cocaine triggers drug-seeking behavior by binding to the dopamine transporter at the synapse [122,123]. A 2021 study revealed that increased zinc concentrations in mice enhance cocaine binding to the dopamine transporter (DAT) protein. Repeated cocaine administration increased zinc concentrations in the caudate putamen (CPu) and nucleus accumbens (NAc). Conversely, low levels of zinc revealed decreased zinc content and cocaine sensitivity in the brain, confirming that zinc plays a role in cocaine-seeking behavior [123]. Additionally, ASIC channels are abundantly expressed in the NAc [122,124]. Previous studies discovered that mice deficient in the ASIC1a gene had increased cocaine-conditioned place preference (CPP); consequently, the effect is abolished when ASIC1a is restored in the NAc [11,125]. More recently, cocaine priming-induced reinstatement and drug-seeking were amplified when ASIC1a was overexpressed in mice NAc [122]. Additionally, our studies have indicated that, at various dosages of cocaine (5, 10, 20, and 30 mg/kg), cocaine drastically reduced acute cocaine-induced motor responses in ASIC1a^-/-^ mice [124]. Behavioral sensitization in chronic cocaine addiction was also prevalent in the same ASIC1a^-/-^ mice, suggesting that ASIC1a plays a role in chronic cocaine-induced behavioral changes [124]. These findings demonstrate that ASIC1a channels partake in both acute and chronic cocaine addiction. Taken together, it is unclear whether zinc-mediated inhibition of ASIC1a would attenuate or stimulate cocaine addiction. Further research is required to investigate the specific therapeutic potential of zinc modulation of ASIC1a channels in cocaine addiction in humans.

Zinc regulation of ASIC1a in certain neurological and psychological disorders has been shown in Figure 2.

### 3.2. ASIC1b

Zinc’s anti-inflammatory qualities from zinc-mediated enhancement/induction of metallothionein are thought to be the reason for its therapeutic effectiveness in the treatment of pain [126], which is significant given inflammation is a primary contributor to the onset of chronic pain, including neuropathic pain [127]. However, the exact role that zinc plays in nociception is unknown. Studies show a diet lacking in zinc enables mice to experience less of the antinociceptive effects of morphine [128]. In addition, zinc was found to inhibit paclitaxel-induced mechanical hypersensitivity, the capsaicin response in DRG neurons, and TRPV1 channels [129]. These are several mechanisms of action that potentially reveal how zinc plays a role in pain sensation. Zinc therapy significantly reduced inflammatory hyperalgesia in a rat model of generated neuropathic pain and levels of the inflammatory biomarker IL-1B and nerve growth factor (NGF) [130].

Consequently, zinc chelation is found to cause hyperalgesia, and a zinc injection leads to pain relief [131,132]. Through TRPV inhibition, zinc may also be used therapeutically to treat chemotherapy-induced peripheral neuropathy [133]. Additionally, the depletion of vesicular zinc in the dorsal root ganglion of mice led to an increase in neuropathic pain due to a lower pain threshold [134]. In mice with fibromyalgia, it was found that when inducing hyperalgesia in both ASIC1b knockout mice and wild-type mice, the knockout mice had a shorter response [132]. Studies also revealed that human and mouse ASIC1b channels exhibit a small sustained current when activated [132,135]. The physiological significance of the sustained current is unclear but has been implicated in persistent pain [136]. Taken together, zinc-mediated inhibition of ASIC1b may play a key role in pain management therapy.

### 3.3. ASIC1a/3

Studies from ASIC knock-out mice have indicated that ASIC1a/3 channels could be one of the main ASIC components within skeletal muscle afferents [137]. ASIC channels play a significant role in detecting protons within sensory muscle neurons, particularly with sensing drops in pH [54,58]. Skeletal muscle afferents are predominantly composed of ASIC heteromers, ASIC1a/2a/3 or ASIC1a/3 [137]. Additional studies have hypothesized that ASIC1a/3 heterotrimers are involved with muscle pain [138]. This study found that PcTx1 significantly inhibited pH 6-evoked currents in ASIC1a/3 heteromeric channels in CHO cells [138]. PcTx1 binds a location on the ASIC1a/3 extracellular domain that controls pH-dependent channel desensitization [138]. With the presence of PcTx1, ASIC1a/3 channels are desensitized at a neutral pH and are, therefore, unable to be opened by acidic pH changes [138]. Thus, targeted inhibition of ASIC1a/3 can be a therapeutic option for patients with activity-induced hyperalgesia [138]. Zinc exhibits dual effects on ASIC1a/3 heterotrimers, causing inhibition at higher concentrations [4]. For example, the pretreatment of zinc of 1–100 µM has an EC_50_ of 24 µM, 100–250 µM has an EC_50_ of 128 µM, and 300 µM has an IC_50_ of 306 µM [4]. Furthermore, it has been shown that systemic zinc administration reduces hyperalgesia during early inflammation by decreasing cytokine IL-beta and growth factor NGF [130]. Therefore, zinc-mediated inhibition of ASIC1a/3 in the DRG of skeletal muscle cells may be a method of treatment for activity-induced pain, and further investigation should be done to explore this relationship.

Furthermore, ASIC1a and ASIC3 channels have a significant expression in the retina and subsequently can form functional ASIC1a/3 channels. In addition, zinc is released into the retina during neurotransmission. Multiple studies have shown that exogenous dietary supplementation of zinc is vital in preventing retinal aging [139,140], age-related macular degeneration [141,142], and maintaining the taurine system [143]. Hence, we hypothesize that zinc may be a modulator of ASIC1a/3 activity under physiological and pathological conditions. Prospective studies are required to investigate the relationship between zinc and ASIC1a/3 channels on the retina and explore zinc’s effects on retinal pathological conditions.

### 3.4. ASIC1a/2b

ASIC1a/2b expression in dorsal horn neurons is upregulated by peripheral inflammation, thereby suggesting a vital role in central pain perception in physiological and pathological circumstances [144]. Similar to homomeric ASIC1a channels, ASIC1a/2b channels are calcium-permeable [55,145]. ASIC1a/2b calcium permeability plays a significant role in mediating neuronal death under various pathological states [55]. When the *ACCN1* (ASIC2) gene is mutated, ASIC1a/2a and ASIC1a/2b heteromers display significantly reduced acidosis-induced calcium rise after an ischemic event [146,147]. Current studies indicate that barium can reduce acid-induced neuronal death by inhibiting ASIC1a/2b [55]. Surprisingly, although ASIC1a homomers resist barium, barium could still profoundly inhibit neuronal death [55]. These results could be due to differences in the expression of ASIC1a/2b channels compared to ASIC1a homomeric channels in the neurons studied. However, this could also be evidence that ASIC1a/2b can be specifically targeted for treatment in neuron populations that are not affected by inhibitors of ASIC1a homomers. No current studies suggest the association between zinc regulation of ASIC1a/2b and tissue acidosis. Since millimolar and nanomolar concentrations of zinc can inhibit ASIC1a/2b as well as ASIC1a, additional research regarding the effects of zinc-mediated inhibition on neuronal cell death under ischemic conditions is necessary to determine whether zinc could be an advantageous therapy.

### 3.5. ASIC1a/2a

ASIC1a-containing channels, such as ASIC1a/2a heterotrimers, are indicated in neuronal injury after an ischemic stroke [49,147]. Although ASIC1a/2a channels are calcium impermeable, in acidic conditions, they can increase calcium levels and contribute to neuronal or axonal degeneration through secondary mechanisms [147,148]. This is true for calcium permeable ASIC1a channels as well. In fact, the activation of VGCC and release of calcium stores in acidotic and ischemic conditions have a higher contribution to the total increase in calcium than the calcium influx through ASIC1a [147]. While the activation of ASIC1a/2a channels may not directly contribute to injury, the greater the number of ASIC1a-containing channels, the greater the magnitude of the acid-induced calcium release during acidosis. Thus, the inhibition of the ASIC1a/2a heterotrimers with nanomolar concentrations of zinc could potentially attenuate the effect of the channel on a neuronal injury during acidotic and ischemic conditions [148,149]. However, as previously mentioned with ASIC1a, current research demonstrates that zinc plays a controversial role in the pathophysiology of ischemic stroke. In one study, the accumulation of zinc in the brains of adult rats was shown to be an independent risk factor for ischemic stroke [116]. In another study on rats, zinc supplementation after ischemia protected the hippocampus from neuronal injury [117]. With these conflicting findings, it is difficult to theorize the effects of zinc modulation of ASIC1a/2a on reducing the likelihood of neuronal degeneration after an ischemic stroke without additional research.

ASIC1a/2a heteromers and ASIC1a homomers are the most commonly expressed ASICs in the neurons of the olfactory bulb and are necessary for odorant stimulation and synaptic transmission [150]. A study proposed that a reduced ASIC function could contribute to the loss of olfactory perception in PD [150]. Additional research on the pathophysiology of ASICs and PD could be beneficial in determining whether micromolar concentrations of zinc could help potentiate ASIC1a/2a and attempt to restore olfactory perception. Currently, studies show that zinc treatment has controversial roles in PD depending on the affinity at which it binds to proteins in different signaling pathways. In one study on Drosophila flies, zinc treatment increased the lifespan and motility of the flies with PD [151]. In another study on mice, zinc supplementation resulted in methamphetamine-induced dopaminergic neuronal loss [94]. Thus, further research is necessary to elucidate if ASIC1a/2a is a channel that zinc will preferentially bind to for the restoration of olfactory perception in patients with PD or if it is more likely to bind to another protein that could result in neurotoxicity.

### 3.6. ASIC2a

ASIC2a homomers are indicated in numerous pathological conditions that affect different parts of the body. While ASIC1a, ASIC2a, and ASIC2b are highly expressed in the CNS, ASIC2a expression is thought to increase susceptibility to temporal lobe epilepsy [59]. There is greater ASIC2a expression post-seizure, and overexpression of ASIC2a hastened the onset of the first epileptic episode as well as increased the occurrence of status epilepticus episodes reaching Racine stage IV [59]. In those with focal brain ischemia due to a stroke, the deletion of ASIC2 channels has a protective effect on hippocampal, cortical, and striatal neurons by decreasing the effects of acidosis-induced injury [147,152]. ASIC2 deletion contributes to this protective effect by deleting heteromeric 1a/2a channels and decreasing the expression of ASIC1a [147]. It also eliminates ASIC2a, and this will result in a reduction of neuronal cell death under pathological conditions [147]. Since zinc helps potentiate ASIC2a currents and could increase seizure susceptibility and acid-induced injury after a stroke, systemic treatment with zinc may result in increased adverse effects for these patients.

In the eyes, ASIC2a is important for retinal function due to its expression in retinal ganglion cells. Zinc, abundant in the retina, can potentiate ASIC2a and consequently protect the eye from light-induced retinal degeneration [153]. Zinc supplementation has been studied to slow the progression of age-related macular degeneration [141,142]. However, in conditions such as glaucoma, in which there is optic nerve injury, retinal ganglion cells cannot regenerate and begin to die. In this circumstance, zinc accumulates in the retina and further prevents the axon regeneration of the retinal ganglion cells [154,155]. Studies propose that zinc chelation in patients with glaucoma could promote the regenerative capacity of the retinal ganglion cells [154,155]. The role of ASIC2a in the progressive damage of the retina after injury to the optic nerve is currently unclear. Further investigation is necessary to elucidate whether the modulation of ASIC2a by zinc plays a role in patients with glaucoma.

### 3.7. ASIC2a/3

ASIC2a/3 is a heteromeric channel found in cardiac dorsal root ganglion neurons [156]. Potentiation of these channels increases the firing of action potentials and could cause persistent angina during myocardial infarction [156]. Previous studies have shown that zinc’s antioxidative and anti-inflammatory nature can protect against myocardial infarction [157], attenuate cardiac remodeling after infarction [158,159], improve the rate of contraction, and increase myocardial flow rate [160]. Although zinc supplementation has proven to be an effective cardioprotective treatment, since it potentiates ASIC2a/3, it should be studied further to determine if it could cause angina as an adverse effect.

### 3.8. ASIC3

ASIC3 is predominantly expressed within nociceptors and is a target of therapy for acid-induced pain [161]. In animal and human studies, selective ASIC3 drug antagonists are effective in relieving acid-evoked pain [161]. Furthermore, data has indicated that ASIC3 is involved in the maintenance of inflammatory pain [162]. One study utilized CFA-treated rats to demonstrate the role of ASIC3 in inflammatory pain [162]. Injection of APETx2, a potent peptide inhibitor of ASIC3, into a CFA-induced inflamed paw of the rat resulted in a complete reversal of mechanical hypersensitivity levels before the CFA treatment, reaching maximal efficacy 30 min post-APETx2 injection [162]. Therefore, inhibition of ASIC3 serves as a powerful therapeutic agent for both acid-induced and inflammatory nociception. Zinc inhibits the peak and sustained components of ASIC3 channels; hence, zinc may play a role in treating chronic pain [9]. Furthermore, ASIC3 is expressed within neuronal cells in the brain and adipocytes, and ASIC3 knockout mice show reduced anxiety levels and enhanced insulin sensitivity [161]. Thus, inhibition of ASIC3, potentially via zinc, may have additional analeptic implications outside of pain management.

ASIC3 has profound expression in the retina, and zinc is released during retinal neurotransmission; therefore, zinc may influence ASIC3 and ASIC1a/3 activity during normal or pathological conditions [4]. As previously stated, multiple studies have shown that exogenous dietary supplementation of zinc is vital in preventing retinal aging [140], age-related macular degeneration [141,142], and maintaining the taurine system [143]. Hence, we hypothesize that zinc may be a modulator of ASIC1a/3 activity under physiological and pathological conditions. Additional research is warranted regarding the zinc regulation on ASIC3 and ASIC1a/3 channels in the retina.

ASIC3 was also discovered to have an influential expression in the bladder. Subsequently, mutations of this channel can cause micturition-affiliated pathophysiological conditions such as urinary incontinence [163]. ASIC3 channels are significantly acid-sensitive and therefore play a vital role in tissue acidosis, nociception, and mechanosensation [52,164]. Knowing that zinc exhibits inhibitory behavior on ASIC3 channels, zinc may consequently play a vital role in tissue acidosis, nociception, and other underlying pathological conditions. A recent study from 2019 discovered that zinc is a protective biomolecule in ischemia/reperfusion injury in various organs [165]. However, no current studies support this relationship between ASIC3 and zinc and warrant future research.

### 3.9. ASIC4

ASIC4, unlike most ASIC channels, does not induce currents when protonated [5]. It is also mainly found intracellularly in endosome-related vacuoles [7]. ASIC4 is hypothesized to be a modulator that downregulates ASIC1a and ASIC3 surface expression [6]. This ultimately reduces the production of currents that trigger acidosis. Furthermore, ASIC4 channels may play a role in reducing fear and anxiety by modulating ASIC1a channels. One study found that knockout mice with ASIC4 mutations demonstrated heightened fear and anxious behavior [47]. Chemo-sensing and mechano-sensing roles of ASIC4 channels are still unclear since they do not form functional heteromeric or homomeric channels [164].

There is a potential linkage between zinc and ASIC4 channels found in zebrafish. Zebrafish ASIC4 (zASIC4) shares characteristics of a cytoplasmic N terminal domain with mammalian ASIC4 that is completely conserved in human ASIC4 channels [166]. zASIC4.1 has a transient component and a sustained component induced by calcium influx. In the study, the administration of 0.5 mM of zinc blocked the sustained component of zASIC4.1 channels but not the transient component, indicating a potential inhibitory modulation by zinc [166,167]. Further research is needed to analyze a possible relationship between zinc and human ASIC4 channels.

### 3.10. ASIC5

Similar to ASIC4, ASIC5 channels are also not activated by protons [168,169]. Recent studies have indicated that ASIC5 is important for type II Unipolar brush cells activity and that disruption of ASIC5 contributes to impaired movement, likely, at least in part, due to altered temporal processing of vestibular input [170]. In fact, ASIC5 has been discovered to be sensitive to bile acids. The closed state of ASIC5 is destabilized in the presence of bile acids, thus activating the channel [169]. It was subsequently named bile acid-sensitive ion channel (BASIC) [168,169]. In addition, physiological concentrations of magnesium and calcium strongly inhibited ASIC5 channels, indicating that extracellular divalent cations stabilize the inactive state of ASIC5 channels [168,169]. Therefore, bile acids and the displacement of extracellular divalent cations activate ASIC5 channels. There is currently no relevant literature discussing the relationship between zinc and ASIC5 channels, but since zinc is also an extracellular divalent cation with a history of effects on other ASIC channels, it is essential to rule out the possible correlation between zinc and ASIC5.

## 4. Conclusions

In conclusion, there has been significant interest in the function of zinc as a modulator of ASICs. ASICs have been implicated in numerous neurological disorders such as ischemic stroke, epilepsy, PD, and AD. Depending on the concentration of zinc and the ASIC isoform, zinc can exert either a stimulatory or inhibitory effect on the ion channels. With further research, zinc has promising potential to provide therapeutic benefits for various neuropathologies through the modulation of ASICs.

## Figures and Tables

**Figure 1 biomolecules-13-00229-f001:**
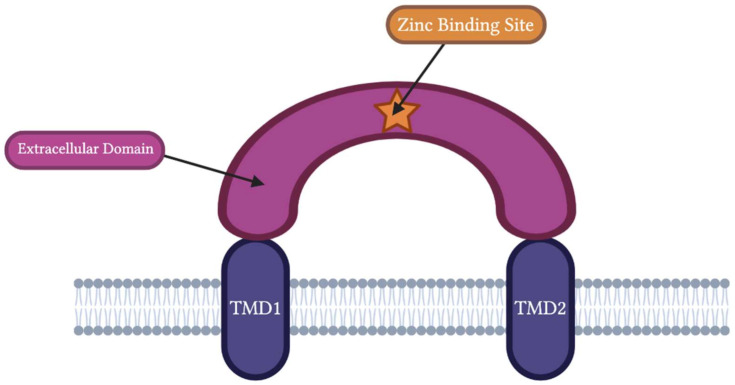
General ASIC structure with the zinc-binding site. Acid-sensing ion channels are composed of two transmembrane domains (TMD1 and TMD2) separated by a large extracellular domain. The *C* and *N* terminal face inside of the membrane. The binding site of zinc is contained within the ASIC extracellular domain. The exact location of zinc’s binding site within this extracellular domain varies depending on the ASIC subtype. Adapted from “Transporters” by BioRender.com (2022 and ac) (https://app.biorender.com/biorender-templates, accessed on 10 December 2022).

**Figure 2 biomolecules-13-00229-f002:**
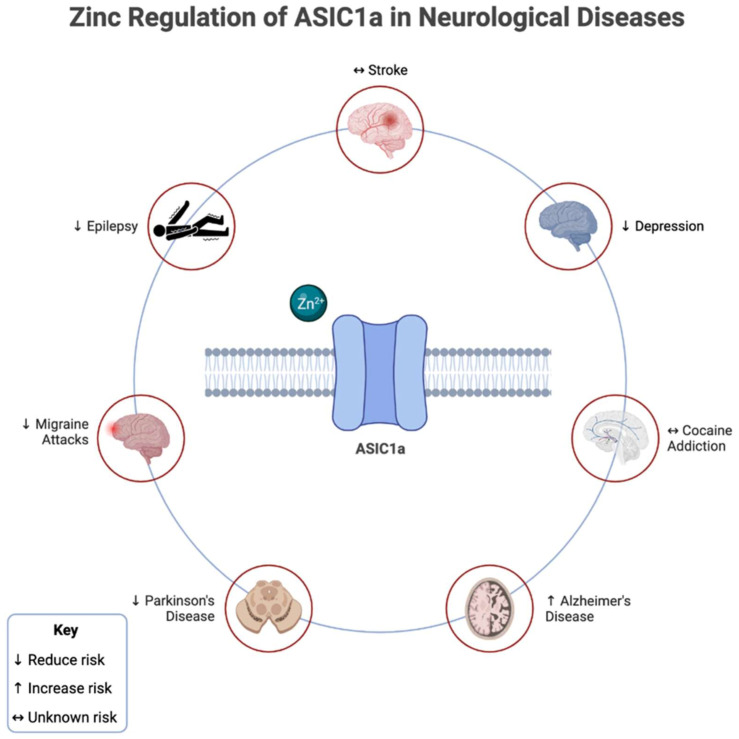
Zinc Regulation of ASIC1a in Neurological Diseases. Zinc-mediated inhibition of ASIC1a channels is theorized to reduce the risk of epilepsy, migraine attacks, Parkinson’s disease, and depression and increase the risk of Alzheimer’s disease. Its effects on cocaine addiction and strokes are unknown based on current literature. Adapted from “Transporters” by BioRender.com (2022) (https://app.biorender.com/biorender-templates, accessed on 28 December 2022).

**Table 1 biomolecules-13-00229-t001:** Summary of zinc’s effects on different ASIC subtypes.

ASIC Subtype	Zinc Binding Site	Zinc Effect	EC_50_/IC_50_	Binding State of Channel	References
ASIC1a	K133^↑^	Inhibitory	IC_50_: 7.0 ± 0.35 nM	Closed	[46]
ASIC1b	C149^↓^	Inhibitory	IC_50_: 36.5 ± 1.5 μM	Closed	[12,51]
ASIC1a/3	H72^↓^, H73^↓^, H83^↓^	Excitatory^*^ [μM] Inhibitory^#^ [1–250 μM] Excitatory^#^ [> 250 μM]	EC_50_^*^: 26 µM EC_50_^#^ [1–100 µM]: 24 µM EC_50_^#^ [100–250 µM]: 128 µM IC_50_^#^ [> 250 µM]: 206 µM	Closed^1,2^ Open^1^	[4]
ASIC1a/2b	Unknown	Inhibitory	Unknown	Open	[50]
ASIC1a/2a	H339^↓,1^, K133^↑,2^	Excitatory [μM] Inhibitory [1–250 μM]	EC_50_: 111 μM IC_50_: 10.04 ± 1.23 nM	Open	[18,46]
ASIC2a	H339^↓^, H162^↓^	Excitatory	EC_50_: 120 μM	Open	[18]
ASIC2a/3	H339^↓^, H162^↓^	Excitatory	Unknown	Open	[18]
ASIC3	Unknown^↓^	Inhibitory	IC_50_: 61 ± 3.2 μM	Closed	[52]

^[]^ The values in brackets represent the zinc concentrations used to determine the subsequent findings.^↑^—High affinity; ^↓^—Low affinity; *—Open state; #—Closed state; ^1^—Excitatory; ^2^—Inhibitory.

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
