# Peer review of "The Role of Zinc in Modulating Acid-Sensing Ion Channel Function"

_biomolecules, 2023, doi:10.3390/biom13020229_

Round 1

Reviewer 1 Report

This review article covers the research field of Zinc and ion channels, in particular, ASICs. It will benifit the readers who are working on either or both fileds. This is a good review. However, I have some comments and wish they will helpful for the author's revision. 

The sub-titles in some sections need to be re-orhanized. For example, the authors use the following sections.   1. Introduction; 2. Zinc’s effects on different types of ASICs; 3 Discussion; 4. Conclusion. However, number#1-3 seems not be logical. I assume the authors have messed up the mumber#2. I suggest the authors change to more consistent titles.

Under the big titles, "Introduction" and "discussion". Is it better to add more detailed sub-titles. I was confused about some sections. For example, 1.4 Zinc and ASICs follow by 2. Zinc’s effects on different types of ASICs.

Is it better to change the "discussion" title to another detailed one? For example, " Zinc and ASICs in diseases" becasue section 3 focus on diseases. 

Is it possible to put the Fig.1 and its legend in the same page so that it will be easier to the readers to read them together.

Author Response

1.The subtitles in some sections need to be reorganized. For example, the authors use the following sections. 1. Introduction; 2. Zinc’s effects on different types of ASICs; 3 Discussion; 4. Conclusion. However, number #1-3 seems to not be logical. I assume the authors have messed up number #2. I suggest the authors change to more consistent titles.

Response:

(1) Great point. Section 1 has been changed as suggested to include a new subtitle for clarity and organization, “1.3. Zinc and Disease.”

(2) Regarding section 2 subtitles, we purposefully placed the ASIC channels in the current order because we wanted all the ASIC2 heteromers to be next to each other to ensure clarity. We also put ASIC1a/2b before ASIC1a/2a so that ASIC1a/2a and ASIC2a would be next to each other.

(3) Great point. We changed “3. Discussion” to “3. Zinc Regulation of ASICs in Neurological Diseases” to be more consistent with the following content discussed.   

2. Under the big titles, "Introduction" and "discussion.” Is it better to add more detailed subtitles? I was confused about some sections. For example, 1.4 Zinc and ASICs follow by 2. Zinc’s effects on different types of ASICs.

Response: Great point. We moved the paragraph under “1.4 Zinc and ASICs” to “1.3 Zinc and Ion Channel Regulation,” and we deleted the title “1.4 Zinc and ASICs” to prevent redundancy and confusion.

3. Is it better to change the "discussion" title to another detailed one? For example, " Zinc and ASICs in diseases" because section 3 focuses on diseases?

Response: Great point. Changed to “Zinc Regulation of ASICs in Neurological and Psychological Diseases.”

4. Is it possible to put the Fig.1 and its legend in the same page so that it will be easier for the readers to read them together?

Response: Changed as suggested.

Reviewer 2 Report

This is a comprehensive review in which the role of zinc in regulating ASICs and disturbance of this regulation in participating development of neurological disorders were summarized. I just have some suggestions for improvement of the readability of this paper.

1.     Abstract:“As both ASICs and zinc concentrations are involved in major neurological diseases”, please modify this sentence to make it clear and accurate. 

2.     The Abstract is too little and did not cover all of the aspects of the review. Besides the background, a summary of the major content reviewed should be included. 

3.     The subtitle “3. Discussion”does not match the content of this part. In this part, the abnormal zinc regulation of ASICs in neurological diseases were summarized and discussed. This part is not a discussion based on the previous parts. Please modify the subtitle.

4.     Page 3, line 91-92, “zinc plays a significant inhibitory role within the prostate gland”, this sentence is hard to understand. What is the inhibitory role? In the following part, the authors continue discussing the zinc disturbance in prostate cancers, this content does not match the subtitle “zinc physiology”. Contents in part 1.2 are suggested to be rearranged to make the intrinsic logic clearer.

5.     Page 5, line 210-211: “how the relationship of these two substances can cause several neurological and psychological diseases ”, please modify this sentence, since “relationship ” cannot cause diseases.

6.     Page 5, line 223, “ASIC-mediated current”, please clarify which ASIC, since zinc may have inhibitory or excitatory effect on different ASICs.

7.     Page 6, line 251-253, please modify this sentence to make the grammar right. “concentrations”can not inhibit ASIC channels, it is zinc to play this role.

8.     It would be nice if the authors provide a figure or table to summarize the regulation of zinc on different ASICs, in which the following issues can be included: ASIC subunit, effect of zinc, effective zinc concentration, pH-dependent, does-dependent, key site for zinc-binding and regulation, etc.

9.     Page 9, line 419, “It is worth nothing that…”, I would suggest the author to modify this comment, since study using HEK293 cells can reveal some common mechanisms in cells and thus is also valuable.

10.  All of the abbreviates must be described in full for the first time. Some of the abbreviated names have not been described in full at all such as TPEN, MPTP.

11.  Page 11: Please modify the following expressions to correct the grammar mistakes: Line 504“Alterations to serotonin levels”; Line 523 “GPR 39 knock mice”; Line 536, 540, “mice with absent ASICs channels/alleles”

12.  Page 12, line 584-591, the content in this part is misleading and confusing, please modify this part.

13.  Page 13, line 597-598, grammar error, please modify this sentence; line 615-616, please delete one“rat”; line 607, metallothionein ( MT ) is group of conjugated proteins which are rich in cysteine, zinc is a metal. Please make sure that this description is accurate.

Author Response

1.Abstract: “As both ASICs and zinc concentrations are involved in major neurological diseases”, please modify this sentence to make it clear and accurate.

Response: This sentence has been removed after modifying the abstract based on point 2 listed below.

2. The Abstract is too little and did not cover all of the aspects of the review. Besides the background, a summary of the major content reviewed should be included.

Response: Thank you for the suggestion. The abstract has been modified to include a summary of the major content discussed in the paper aside from the background. It is now as follows: “Acid-sensing ion channels (ASICs) are proton-gated, voltage-independent sodium channels that are widely expressed throughout the central and peripheral nervous systems. Zinc, an important trace metal in the body, is involved in numerous physiological functions, with neurotransmission being of note. Zinc has been implicated in the modulation of ASICs by binding to specific sites on these channels and exerting either stimulatory or inhibitory effects depending on the ASIC subtype. ASICs have been linked to several neurological disorders such as Alzheimer’s disease, Parkinson’s disease, ischemic stroke and epilepsy. Different ASIC isoforms contribute to the persistence of each of these neurological disorders. By understanding how various zinc concentrations can modulate specific ASIC subtypes, zinc may be an important therapeutic target for certain neurological diseases.”

3.The subtitle “3. Discussion” does not match the content of this part. In this part, the abnormal zinc regulation of ASICs in neurological diseases were summarized and discussed. This part is not a discussion based on the previous parts. Please modify the subtitle.

Response: Changed as suggested to “Zinc Regulation of ASICs in Neurological Diseases.”

4. Page 3, line 91-92, “zinc plays a significant inhibitory role within the prostate gland,” this sentence is hard to understand. What is the inhibitory role? In the following part, the authors continue discussing the zinc disturbance in prostate cancers, this content does not match the subtitle “zinc physiology.” Contents in part 1.2 are suggested to be rearranged to make the intrinsic logic clearer.

Response: Zinc plays a role in inhibiting growth within the prostate gland. To further clarify zinc’s role in the prostate gland, the sentence has been changed to “In addition, zinc plays a significant role in inhibiting growth within the prostate gland.” Furthermore, in order to make the contents in part 1.2 clearer, this section has been split into two parts: 1.2 Zinc Physiology and 1.3 Zinc and Disease. Discussion of zinc and prostate cancer as well as zinc’s involvement in neurological disease has been moved to section 1.3 Zinc and Disease.

5. Page 5, line 210-211: “how the relationship of these two substances can cause several neurological and psychological diseases,” please modify this sentence, since “relationship” cannot cause diseases.

Response: Great point. Sentence modified to “Thus, the next section of the review goes into detail about the relationship between zinc and ASICs and how the relationship of these two substances correlates with several neurological and psychological diseases.” 

6. Page 5, line 223, “ASIC-mediated current,” please clarify which ASIC, since zinc may have inhibitory or excitatory effect on different ASICs.

Response: Changed to “ASIC1a-mediated currents.”

7. Page 6, line 251-253, please modify this sentence to make the grammar right. “concentrations”can not inhibit ASIC channels, it is zinc to play this role.

Response: Changed to “Zinc concentrations of 1 or 3 μM.”

8. It would be nice if the authors provide a figure or table to summarize the regulation of zinc on different ASICs, in which the following issues can be included: ASIC subunit, effect of zinc, effective zinc concentration, pH-dependent, dose-dependent, key site for zinc-binding and regulation, etc.

Response: Thank you for the suggestion. We have added a table addressing the following characteristics of each ASIC subunit: binding site and affinity, the effect of zinc, EC50/IC50, and the state of the channel during binding. This summary table will be at the end of section “2. Zinc’s effects on different types of ASICs.”

9. Page 9, line 419, “It is worth noting that…”, I would suggest the author to modify this comment, since study using HEK293 cells can reveal some common mechanisms in cells and thus is also valuable.

Response: Changed to “The corresponding study used HEK 293 cells, so prospective research is needed on neuronal cell lines to study the neuroprotection of different concentrations of extracellular zinc.”

10. All of the abbreviates must be described in full for the first time. Some of the abbreviated names have not been described in full at all such as TPEN, MPTP.

Response: Changed as suggested.

11. Page 11: Please modify the following expressions to correct the grammar mistakes: Line 504“Alterations to serotonin levels”; Line 523 “GPR 39 knock mice”; Line 536, 540, “mice with absent ASICs channels/alleles”

Response: Line 504 has been changed to “A key contributing factor to depression may be alterations to serotonin levels in the brain.” Line 523 has been changed to “Consequently, GPR39 knock-out mice are resistant to traditional antidepressants.” Lines 536-540 have been changed to “Using the forced swim test and tail suspension test, mice with absent ASIC1a channels displayed antidepressant-like findings in comparison to mice with normal ASIC1a. Administration of PcTx1 and amiloride also produced antidepressant-like effects. Overall, findings from this study reveal that ASIC1a contributes to depression in mice [110].”

12. Page 12, lines 584-591, the content in this part is misleading and confusing, please modify this part.

Response:

(1) Lines 594-600 have been changed to “Cocaine triggers drug-seeking behavior by binding to the dopamine transporter at the synapse [122,123]. A 2021 study revealed that increased zinc concentrations in mice enhance cocaine-binding to the dopamine transporter (DAT) protein. Repeated cocaine administration led to increased zinc concentrations in the caudate putamen (CPu) and nucleus accumbens (NAc). Conversely, low levels of zinc revealed decreased zinc content and cocaine sensitivity in the brain, confirming that zinc plays a role in cocaine-seeking behavior [123].”

(2) Lines 610-613 have also been modified to “Taken together, it is unclear whether zinc-mediated inhibition of ASIC1a would attenuate or stimulation cocaine addiction. Further research is required to investigate the specific therapeutic potential of zinc modulation of ASIC1a channels in cocaine addiction in humans” to clarify the main discussion.

13. Page 13, line 597-598, grammar error, please modify this sentence; line 615-616, please delete one “rat”; line 607, metallothionein (MT) is group of conjugated proteins which are rich in cysteine, zinc is a metal. Please make sure that this description is accurate.

Response:

(1) Changed as suggested.

(2) Changed to “Zinc's anti-inflammatory qualities from zinc-mediated enhancement/induction of metallothionein are thought to be the reason for its therapeutic effectiveness in the treatment of pain [126], which is significant given inflammation is a primary contributor to the onset of chronic pain including such neuropathic pain [127].”

Reviewer 3 Report

The review by Sun et al., “The Role of Zinc in Modulating Acid-sensing Ion Channel

Function” has summarized zinc modulates acid-sensing ion channels are widely expressed throughout the central and peripheral nervous systems and have been linked to several neurological disorders. This paper was well written, and a few minor comments needed to be considered before publication.

A few concerns/comments needed to be explained/modified:

1.     line 85: “the most abundant trace metal” It is recommended to add specific values to be more convincing.

2.     Line 86: In adults, only about 1.5% of zinc is present in the brain, so the expression of “majority” is inappropriate.

3.     Line 406: “Epilepsy is a neurological disorder characterized by seizures.” The statement is too general, epilepsy is characterized by chronic brain seizures caused by abnormal, excessive or synchronized neuronal activity.

4.     Line 427-429: “A 2020 randomized 8-week clinical trial indicated that zinc supplementation reduces the frequency of migraine attacks in patients.” More detailed data are necessary, how the level of zinc supplementation (and in what form) can reduce the frequency of migraine attacks?

5.     Line 479-480: “Zinc pretreatment reverses the aforementioned phenomena by increasing metallothionein expression in vitro.” What is the relationship between metallothionein expression and dopaminergic cell death? More detail please.

6.     Line 514-515: “zinc modulates a variety of ligand- and voltage-gated ion channels.” Please list the specific ion channels.

7.     Line 581: “zinc administration or chelation is an unreliable treatment” It is mentioned above that zinc supplementation can improve the neurological deficit, such information is imprecise and will be unclear for a reader.

8.     ASIC1a is intimately linked to numerous neurological disorders, it would be better to provide a related figure.

Author Response

1.Line 85: “the most abundant trace metal” It is recommended to add specific values to be more convincing.

Response: Great point. Additional information has been added as suggested, “The human body contains approximately 2-4 grams of zinc, and the majority of zinc is distributed within the testes, muscles, liver, and prostate.”

2. Line 86: In adults, only about 1.5% of zinc is present in the brain, so the expression of “majority” is inappropriate.

Response: Thank you for pointing this discrepancy out. The comment suggesting the majority of zinc is in the brain has been deleted.

3. Line 406: “Epilepsy is a neurological disorder characterized by seizures.” The statement is too general, epilepsy is characterized by chronic brain seizures caused by abnormal, excessive, or synchronized neuronal activity.

Response: Changed to “Epilepsy is a neurological disorder characterized by seizures due to abnormal, excessive, or synchronized neuronal activity.”

4. Line 427-429: “A 2020 randomized 8-week clinical trial indicated that zinc supplementation reduces the frequency of migraine attacks in patients.” More detailed data are necessary, how the level of zinc supplementation (and in what form) can reduce the frequency of migraine attacks?

Response: Changed to “A 2020 randomized 8-week clinical trial revealed that 220 mg of zinc sulfate per day reduced the frequency of migraine attacks in comparison to the placebo group. However, other factors such as headache severity, migraine duration, and presence of auras were not affected by zinc supplementation.”

5. Line 479-480: “Zinc pretreatment reverses the aforementioned phenomena by increasing metallothionein expression in vitro.” What is the relationship between metallothionein expression and dopaminergic cell death? More detail please.

Response: Changed to “Zinc pretreatment reverses the aforementioned phenomena by increasing metallothionein expression in vitro, attenuating the accumulation of ROS in neurons [92,93]. By pretreating the cells with 50 μM of zinc chloride, methamphetamine-induced expression of α-synuclein was significantly reduced [92].”

6. Line 514-515: “zinc modulates a variety of ligand- and voltage-gated ion channels.” Please list the specific ion channels.

Response: Changed to “Zinc can act as an inhibitory neuromodulator of NMDA channels, a major pharmacotherapeutic target in depression, [100] or it can act as a neurotransmitter [101].”

7. Line 581: “zinc administration or chelation is an unreliable treatment” It is mentioned above that zinc supplementation can improve the neurological deficit, such information is imprecise and will be unclear for a reader.

Response: This is a great point. The sentence “However, despite extensive research on zinc modulation and ASIC1a channels, with conflicting data on zinc’s role in ischemic strokes, zinc is an unreliable treatment and requires further research to clarify zinc’s significance in the human brain during an ischemic stroke.”  has been removed. It has been changed to “Within the current literature, the effect of zinc on ischemic stroke is inconclusive, as different studies have seen the neuroprotective effects of both zinc chelation and administration. However, further research on the effect of zinc chelation and administration in the early versus late stages of ischemic stroke may ascertain whether the neuroprotective effects of zinc chelation are time dependent and/or superior to the effects of zinc administration.”

8. ASIC1a is intimately linked to numerous neurological disorders, it would be better to provide a related figure.

Response: This is a great idea. Figure 2 titled “Zinc Regulation of ASIC1a in Neurological Diseases” along with a description has been added at the end of section “3.1 ASIC1a.”